# Associations of Food and Nutrient Intake with Serum Hepcidin and the Risk of Gestational Iron-Deficiency Anemia among Pregnant Women: A Population-Based Study

**DOI:** 10.3390/nu13103501

**Published:** 2021-10-03

**Authors:** Noor Rohmah Mayasari, Chyi-Huey Bai, Tzu-Yu Hu, Jane C.-J. Chao, Yi Chun Chen, Ya Li Huang, Fan-Fen Wang, Alexey A. Tinkov, Anatoly V. Skalny, Jung-Su Chang

**Affiliations:** 1School of Nutrition and Health Sciences, College of Nutrition, Taipei Medical University, Taipei 110, Taiwan; rohmah.noor29@gmail.com (N.R.M.); ma07106003@tmu.edu.tw (T.-Y.H.); chenjui@tmu.edu.tw (J.C.-J.C.); yichun@tmu.edu.tw (Y.C.C.); 2School of Public Health, College of Public Health, Taipei Medical University, Taipei 110, Taiwan; baich@tmu.edu.tw (C.-H.B.); ylhuang@tmu.edu.tw (Y.L.H.); 3Department of Public Health, School of Medicine, College of Medicine, Taipei Medical University, Taipei 110, Taiwan; 4Department of Internal Medicine, Yangming Branch, Taipei City Hospital, Taipei 11146, Taiwan; dag90@tpech.gov.tw; 5Laboratory of Ecobiomonitoring and Quality Control, Yaroslavl State University, 150003 Yaroslavl, Russia; tinkov.a.a@gmail.com; 6Laboratory of Molecular Dietetics, IM Sechenov First Moscow State Medical University (Sechenov University), 119435 Moscow, Russia; skalnylab@gmail.com; 7Laboratory of Medical Elementology, KG Razumovsky Moscow State University of Technologies and Management, 109004 Moscow, Russia; 8Graduate Institute of Metabolism and Obesity Sciences, College of Nutrition, Taipei Medical University, Taipei 110, Taiwan; 9Nutrition Research Center, Taipei Medical University Hospital, Taipei 110, Taiwan; 10Chinese Taipei Society for the Study of Obesity (CTSSO), Taipei 110, Taiwan

**Keywords:** foods and nutrients, hepcidin, iron-deficiency anemia, pregnancy, Nationwide Nutrition and Health Survey in pregnant women

## Abstract

Hepcidin is a regulator of iron metabolism. Diet affects the body’s iron status, but how it influences hepcidin concentrations and the risk of gestational iron-deficiency anemia (IDA) remains unclear. We investigated relationships of food and nutrient intake with serum hepcidin levels in relation to the iron status at a population scale. A retrospective cross-sectional study was conducted based on data obtained from the Nationwide Nutrition and Health Survey in pregnant women, Taiwan (2017~2020). In total, 1430 pregnant women aged 20~45 years with a singleton pregnancy were included. Data from blood biochemistry, 24-h dietary recall, and a food frequency questionnaire were collected during a prenatal checkup. Adjusted multivariate linear and logistic regression analyses were employed to measure the beta coefficient (ß) and 95% confidence interval (CI) of serum hepcidin and the odds ratio (OR) of IDA. In IDA women, serum hepcidin levels were positively correlated with the intake frequency of Chinese dim sum and related foods (β = 0.037 (95% CI = 0.015~0.058), *p* = 0.001) and dark leafy vegetables (β = 0.013 (0.001~0.025), *p* = 0.040), but they were negatively correlated with noodles and related products (β = −0.022 (−0.043~−0.001), *p* = 0.038). An adjusted multivariate logistic regression analysis showed that dietary protein [OR: 0.990 (0.981~1.000), *p* = 0.041], total fiber [OR: 0.975 (0.953~0.998), *p* = 0.031], and rice/rice porridge [OR: 1.007 (1.00~1.014), *p* = 0.041] predicted gestational IDA. Total carbohydrates [OR: 1.003 (1.000~1.006), *p* = 0.036], proteins [OR: 0.992 (0.985~0.999), *p* = 0.028], gourds/shoots/root vegetables [OR: 1.007 (0.092~1.010), *p* = 0.005], and to a lesser extent, savory and sweet glutinous rice products [OR: 0.069 (0.937~1.002), *p* = 0.067] and dark leafy vegetables [OR: 1.005 (0.999~1.011), *p* = 0.088] predicted IDA. The risk of IDA due to vegetable consumption decreased with an increasing vitamin C intake (*p* for trend = 0.024). Carbohydrates and vegetables may affect the gestational iron status through influencing hepcidin levels. Vitamin C may lower the risk of gestational IDA due to high vegetable consumption.

## 1. Introduction

Iron-deficiency (ID) anemia (IDA) is recognized as the most widespread nutritional deficiency affecting 56 million pregnant women worldwide [1]. Globally, trends and prevalences of gestational anemia vary significantly across regions and countries [2]. The prevalence of anemia among pregnant women ranges from 56% in Central and West Africa and 52% in South Asia to 22% in high-income countries [2]. In North America, the prevalence of gestational ID is very low (7.3%) in the first trimester but significantly increases in the second (23.7%) and third trimesters (39.2%) [3]. This corresponds to a progressive rise in gestational IDA in the first, second, and third trimesters: 2.7%, 2.2%, and 10.8%, respectively [3]. However, a recent survey showed an increasing trend of gestational ID (23.8%) in the first trimester in 2021 [4]. Gestational IDA is associated with the risk of maternal mortality and both cognitive and behavioral abnormalities of neonates [5,6].

Iron bioavailability in pregnancy is controlled by the hepcidin hormone. Hepcidin, a negative master iron regulator, controls iron homeostasis through proteolytic degradation of ferroportin, the only known mammalian iron exporter [7]. Currently, how the hepcidin level is regulated during pregnancy remains unclear. Outside of pregnancy, hepcidin is downregulated by low body iron storage, hypoxia, and anemia (e.g., erythropoietic activity), and it is upregulated by inflammation or high iron stores [7,8]. In healthy pregnancies, maternal hepcidin concentrations are gradually downregulated in the second and third trimesters, and they are nearly undetectable by late pregnancy [7,9,10,11]. Decreasing circulating hepcidin levels during the gestational stages indicates maternal iron deprivation, and it is vital for the maintenance of proper maternal and embryonic iron homeostasis [12]. Low levels of serum hepcidin enhances maternal dietary iron absorption and the transfer of dietary iron across the placenta, resulting in enhancement of the neonatal iron status [13]. In contrast, high gestational hepcidin levels decrease dietary iron absorption and impair the efficacy of maternal iron supplementation, as well as the transfer of maternal iron to the fetus, leading to restricted iron bioavailability of the fetus. Overall, low levels of maternal hepcidin may allow a supply of iron into the circulation-placenta-neonate axis and facilitate iron endowment to the fetus [7].

A few studies investigated the relationship of diet with the hepcidin-iron axis [14,15,16,17], but none of them focused on pregnancy. Foods such as vegetables and carbohydrates can potentially affect hepcidin levels [14,16,17,18], but results are inconsistent. Vegetables such as dark leafy vegetables and bamboo shoots have an iron-inhibitory effect, presumably due to phytate- or phenolic compound-mediated iron-chelating effects [19,20]. A study found that adolescent girls who had the highest intake of green leafy vegetables (e.g., amaranth) had significantly higher levels of serum hepcidin and ferritin compared to those with the lowest intake [16]. Administration of *T. occidentalis* leaves led to modest increases in hemoglobin (Hb) and hepatic iron levels and hepcidin messenger (m)RNA expression in animals with a diet-induced ID [14]. In contrast, supplementation with moringa leaf to rats with diet-induced ID resulted in increased serum iron levels but decreased hepatic hepcidin and ferroportin expressions [17]. Vegetables are also polyphenol-rich foods, and polyphenols may regulate hepcidin levels. For example, studies showed that quercetin [21] and myricetin [22] suppress hepcidin synthesis via the bone morphogenetic protein 6 (BMP6)/mothers against decapentaplegic homolog 4 (SMAD4) signaling pathways. In contrast, an animal study showed that a quercetin injection decreased the non-heme iron absorption rate, possibly through downregulating ferroportin, but it enhanced hepcidin synthesis [23]. Carbohydrate-containing foods such as cereals and brown/milled rice may inhibit iron absorption due to the presence of phytates and fiber [24,25]. Ascorbic acid (vitamin C) can prevent the iron-chelating effects of dietary fiber or phenolic compounds in vegetables and carbohydrate-rich foods, and it can also stimulate iron uptake by inhibiting hepcidin expression [26].

ID is still the most common cause of anemia in pregnant women. Currently, how foods and nutrients affect maternal hepcidin levels and the risk of gestational IDA remain largely unknown. By studying a nationwide representative population, the broad aim of this study was to investigate the relationships of food and nutrient intake with serum hepcidin and the risk of gestational IDA at a population scale among pregnant women in Taiwan.

## 2. Materials and Methods

### 2.1. Study Design and Population

This study was a retrospective cross-sectional study based on data obtained from the Nationwide Nutrition and Health Survey in pregnant women, which was conducted in Taiwan in 2017~2019 (Pregnant NAHSIT 2017~2019). Stratified probability sampling was used to stratify the sample population into four strata: northern, central, southern, and eastern regions. For each stratum, at least two medical facilities were selected according to the number of prenatal examinations each year. Pregnant women were recruited at the prenatal care center, and they had to meet all of the following criteria to be eligible for the study: (1) aged 15 years or older, (2) with a Taiwanese residency and able to speak fluent Chinese or Taiwanese, (3) had received a maternal health handbook, and (4) agreed to participate and provided a copy of signed written informed consent (*N* = 1502). In Taiwan, prenatal care checkups are based on recommendations in the maternal health handbook in the interval of 13~17 weeks for the first trimester, 22~26 weeks for the second trimester, and >29 weeks for the third trimester. In this study, 72 participants were excluded due to being underage (*n* = 7), having multiple pregnancies (*n* = 33), and lacking information on blood biochemistry or dietary data (*n* = 32). In total, 1430 respondents were included in the analysis. All participants provided written informed consent. The study was approved by the Institutional Review Board of Taipei Medical University (no. TMU-JIRB N201707039) and was therefore performed in accordance with ethical standards laid down in the *Declaration of Helsinki*.

### 2.2. Data Collection

In brief, pregnant NAHSIT 2017–2019 included three components: (1) a self-reported questionnaire consisting of socioeconomic data, health history, and use of prenatal supplements; (2) a face-to-face interview to obtain habitual dietary intake; and (3) blood biochemistry. A self-reported baseline questionnaire included information such as age, pre-pregnancy body-mass index (pBMI), region, trimester, parity, and reported use of prenatal dietary supplements (e.g., multivitamins, vitamin B12, folate, and iron). The pBMI (kg/m^2^) was calculated from self-reported pre-pregnancy body weight and body height. Dietary data were collected by a 24-h recall and food frequency questionnaire (FFQ) during face-to-face interviews by an experienced dietitian. Blood samples were collected during a prenatal visit to assess the iron status.

### 2.3. 24-h Dietary Recall and FFQ

Participants were required to recall all foods or beverages consumed in the past 24 h, and portion sizes were estimated by an experienced dietitian using food models, picture aids, and household utensils. Detailed information on the food consumed (e.g., cooking methods, type of meal, mealtime, snacking, and place) was also recorded. Nutrient intake was calculated using the online software Cofit Pro (Cofit Healthcare, Taipei, Taiwan) based on the Taiwan Food Nutrient Database.

The interviewer-administrated FFQ developed in this study consisted of 59 food items. The FFQ was modified from the NAHSIT Taiwanese FFQ, which originally consisted of 64 food items [27]. In this study, food items were assembled into 33 food groups based on the similarity of food characteristics and nutritional contents (Appendix A). Similar foods were listed close to each other to prevent redundant collection [27]. Participants were asked to recall their usual frequency of each food item during the past month. The frequency of consumption was reported as how often each food was consumed: (1) how many times per day, (2) how many times per week, or (3) how many times per month. The mean consumption of each food group was presented as times/month. A validation study showed that the Taiwanese FFQ for Chinese-speaking people in Taiwan is reproducible and provides a useful measure of habitual intake over a 6-month period [27]. The FFQ also exhibited acceptable reproducibility and validity in assessing most food and nutrient intake levels among pregnant women in China [28].

Initially, 33 food groups were included in the statistical analysis. Since the broad aim of this study was to investigate foods that may influence levels of hepcidin, a master iron regulator, we first performed a multivariate linear regression analysis to investigate the relationship between individual food groups and serum hepcidin levels. Fifteen food items were removed from further analysis due to having no correlation with serum hepcidin levels. The 18 food groups selected for the further investigation included rice/rice porridge/related products; noodles/related products; breakfast cereals/oats/related products; Chinese dim sum/related foods; sweet buns/bean buns/related products; savory and sweet glutinous rice; light-colored vegetables; dark leafy vegetables; gourds and root vegetables; legumes; total vegetables; fresh fruits; 100% pure fruit juice/vegetable juice; canned fruits/dried fruits/jam; fish/shellfish/seafood; roe and processed seafood products; poultry meat; and liver/organs/blood products (Table 2). The FFQ provides the number of frequencies of food intake each month. In this study, the FFQ could not provide nutrient intake estimation. We used a 24-h recall to estimate nutrient intake.

### 2.4. Laboratory Examination

All participants underwent blood collection by venipuncture at a medical facility. Hemoglobin (Hb) was measured by a complete blood cell count using a hematology analyzer (Sysmex, Kobe, Japan). Serum ferritin was measured by a chemiluminescence immunoassay by Beckman Coulter Unicel DxC 800 (Beckman Coulter, Brea, CA, USA). Transferrin saturation (TS) was calculated by dividing serum iron by the total iron-binding capacity (TIBC). Serum iron was measured by a ferrozine-based colorimetric assay and TIBC by an immunoturbidimetric method with a Beckman Coulter Unicel DxC 800 (Beckman Coulter). Serum hepcidin levels were measured with a human hepcidin DuoSet enzyme-linked immunosorbent assay (ELISA) (R&D Systems, Minneapolis, MN, USA), according to the manufacturer’s instructions. Variation in the intra- and inter-assay coefficients of serum hepcidin was <10%.

### 2.5. Definitions

According to the World Health Organization (WHO), ID is defined as TS of <16% [29] and serum ferritin of <15 ng/mL [30]. The Center of Disease Control and Prevention (CDC) defines Hb levels of <11.0 g/dL in the first and third trimesters of pregnancy and <10.5 g/dL in the second trimester as anemia [31]. Gestational IDA is defined as (1) an Hb level of <11 g/dL in the first and third trimesters and <10.5 g/dL in the second trimester, (2) TS of <16%, and (3) serum ferritin of <15 ng/mL. Trimesters (Ts) are defined according to guidelines of the Ministry of Health and Welfare, Taiwan: (1) T1 as the first 17 weeks of pregnancy following the last normal menstrual period, (2) T2 as weeks 18~28, and (3) T3 as weeks 29~40.

### 2.6. Statistical Analysis

Serum hepcidin levels were stratified into quartiles (Qs), with Q1 being the lowest and Q4 being the highest values. Continuous variables are presented as the mean ± standard deviation (SD), and categorical variables are presented as a number (percentage). Since our data provided a sufficiently large sample population, the linear regression required no assumption of a normal distribution [32]. Total energy intake using a residual method was adjusted for in this study [33]. A general linear model and Chi-squared test were used to analyze the *p* for trend between variables for continuous data and categorical data, respectively. Adjusted multivariate linear and logistic regression analyses were employed to measure the beta coefficient (ß) and 95% confidence interval (CI) of serum hepcidin and the odds ratio (OR) of IDA, using nutrient intake levels and food groups as predictors. The significance level for the *p* for trend and *p* value was defined as <0.05. Data were analyzed using SPSS vers. 21 (IBM, Armonk, NY, USA) and GraphPad Prism 5 (GraphPad Software, San Diego, CA, USA).

## 3. Results

### 3.1. Maternal Baseline Characteristics Stratified by Quartiles (Qs) of Serum Hepcidin Levels

Table 1 shows baseline characteristics of the study participants, according to serum hepcidin levels. Compared to those with the lowest (Q1) serum hepcidin levels, pregnant women with the highest serum hepcidin (Q4) were older, heavier, had the highest proportions of living in the northern area of Taiwan (38.1%), a primiparous rate (59.7%), in trimester 1 (49.9%), and used total prenatal supplements (90.9%) and folate supplements (55.7%) (all *p* for trend <0.01) (Table 1). Blood biochemical analyses showed that serum hepcidin levels were positively correlated with all iron biomarkers of Hb, %TS, and serum ferritin (all *p* for trend <0.001). In contrast, serum hepcidin levels were negatively correlated with the prevalence of anemia (16%), IDA (0%), and ID (0.3%), except for the prevalence of non-IDA anemia (16%), which was positively correlated with serum hepcidin levels (all *p* for trend <0.001) (Table 1).

### 3.2. Associations between Maternal Nutrient/Food Intake and Serum Hepcidin Levels

We next investigated relationships between food and nutrient intake and serum hepcidin levels among pregnant women. Table 2 shows that serum hepcidin levels were positively correlated with intake levels of dietary fiber (Q1: 1.5 ± 8.6 g vs. Q4: 3.0 ± 9.4 g), vitamin C (Q1: 67.7 ± 100.3 g vs. Q4: 86.0 ± 145.2 g), 100% pure fruit juice or fruit/vegetable juice (Q1: 5.4 ± 8.6 times/month vs. Q4: 7.3 ± 12.9 times/month), and canned fruits/dried fruits/jam (Q1: 2.3 ± 9.6 times/month vs. Q4: 4.4 ± 14.5 times/month) (all *p* for trend <0.05). In contrast, negative linear trends were found between serum hepcidin levels and consumption frequencies of breakfast cereals/oats/related products (Q1: 2.9 ± 9.3 times/month vs. Q4: 1.4 ± 8.5 times/month), sweet buns/bean buns/related products (Q1: 3.2 ± 6.2 times/month vs. Q4: 2.1 ± 5.1 times/month), light-colored vegetables (Q1: 23.4 ± 21.3 times/month vs. Q4: 21.2 ± 19.2 times/month), gourds/shoots/root vegetables (Q1: 29.3 ± 26.2 times/month vs. Q4: 25.4 ± 22.8 times/month), legume*s* (Q1: 4.0 ±7.9 times/month vs. Q4: 2.7 ± 4.8 times/month), and fish/shellfish/seafood (Q1: 20.9 ± 20.1 times/month vs. Q4: 19.1 ± 18.0 times/month) (all *p* for trend <0.05) (Table 2).

### 3.3. Potential Foods and Nutrients That Affect Serum Hepcidin Levels

We next performed a multivariate linear regression analysis to investigate the predictive effects of nutrients and foods on serum hepcidin levels stratified by the iron status (Table 3). For all women, serum hepcidin levels were positively associated with intake frequency levels of savory/sweet glutinous rice products (β = 0.677 (95% CI: 0.330~1.024), *p* < 0.001), canned fruits/dried fruits/jam (β = 0.157 (0.009~0.304), *p* = 0.037), and liver/organs/blood products (β = 0.332 (0.139~0.526), *p* = 0.001) but negatively correlated with breakfast cereals/oats/related products (β = −0.187 (−0.370~−0.003), *p* = 0.046), gourds/shoots/root vegetables (β = −0.071 (−0.133~−0.010), *p* = 0.022), and to a lesser extent, light-colored vegetables (β = −0.075 (−0.150~0.001), *p* = 0.052), after adjusting for covariates (age, pBMI, region, parity, total supplement use, and trimester) (Table 3: all women).

Among non-IDA pregnant women, intake frequency levels of savory/sweet glutinous rice products (β = 0.662 (0.218~1.107), *p* = 0.004), roe/processed seafood products (β = 1.223 (0.0.321~2.124), *p* = 0.008), and liver/organs/blood products (β = 0.489 (0.189~0.790), *p* = 0.001) were positively correlated with serum hepcidin levels, but intake of breakfast cereals/oats/related products (β = −0.295 (−0.567~−0.023), *p* = 0.034) was negatively correlated with serum hepcidin levels (Table 3: non-IDA). For ID women, the intake frequency of dark leafy vegetables (β = 0.012 (0.000~0.023), *p* = 0.006) was positively correlated with serum hepcidin levels (Table 3: ID). Among IDA women, serum hepcidin levels were positively associated with the intake frequency of Chinese dim sum/related foods (β = 0.037 (0.015~0.058), *p* = 0.001) and dark leafy vegetables (β = 0.013 (0.001~0.025), *p* = 0.040), but negatively correlated with noodles/related products (β = −0.022 (−0.043~−0.001), *p* = 0.038) (Table 3: IDA).

### 3.4. Predictive Effects of Nutrients and Food Groups on Gestational ID and IDA

Figure 1 shows that after adjusting for age, pBMI, region, parity, total supplement use, and trimester, protein [OR: 0.990 (0.981~1.000), *p* = 0.041], total dietary fiber [OR: 0.975 (0.953~0.998), *p* = 0.031], rice/rice porridge [OR: 1.007 (1.00~1.014), *p* = 0.041], and to a lesser extent, dietary iron [OR: 0.970 (0.953~0.098), *p* = 0.073] independently predicted gestational IDA. Factors independently associated with ID were total carbohydrates [OR: 1.003 (1.000~1.006), *p* = 0.036], proteins [OR: 0.992 (0.985~0.999), *p* = 0.028], gourds/shoots/root vegetables [OR: 1.007 (0.092~1.010), *p* = 0.005], and borderline associations were observed with savory/sweet glutinous rice products [OR: 0.069 (0.937~1.002), *p* = 0.067] and dark leafy vegetables [OR: 1.005 (0.999~1.011), *p* = 0.088], after adjusting for covariates (age, pBMI, region, parity, total supplement use (%), and trimester).

Figure 2A shows that individuals with the highest intake levels of dark leafy vegetables (T3) and the lowest intake levels of vitamin C (T1) had the highest risk for IDA, and the risk decreased with increasing vitamin C consumption; OR = 2.47 (1.32~4.6), 1.17 (0.63~2.18), and 1.31 (0.70~2.44) for T1, T2, and T3, respectively (*p* for trend = 0.024). A similar relationship was observed for individuals with the highest intake levels of gourds/shoots/root vegetables (T3) and vitamin C; OR = 1.92 (1.00~3.68), 1.23 (0.65~2.30), 1.31 (0.68~2.50) for vitamin C at T1, T2, and T3, respectively (Figure 2B).

## 4. Discussion

Hepcidin is the master regulator of iron homeostasis, but its regulation during pregnancy remains unclear. As far as we are aware, this is among the first study to investigate associations of intake of foods and nutrients with hepcidin levels. Using cross-sectional data from 1456 pregnant women who participated in the Nationwide Nutrition and Health Survey in pregnant women in Taiwan (2017~2020) (Pregnant NAHSIT 2017~2020), we found that serum hepcidin was inversely correlated with trimester and the prevalences of ID and IDA. This suggests that as pregnancy progress, serum hepcidin levels decrease as circulating iron cannot meet the demand of erythropoiesis. Low hepcidin levels can enhance dietary iron absorption and iron bioavailability across the placenta to support fetal development. Food-hepcidin relationships seem to be dependent on the iron status of pregnant women. Total carbohydrates, carbohydrate-rich foods (rice/rice porridge), and vegetables (dark leafy vegetables, and gourds/shoots/root vegetables) predicted the risk of ID or IDA. In contrast, dietary protein, total dietary fiber, and to a lesser extent, dietary iron protected against gestational IDA. Increased intake of vitamin C may lower the risk of IDA associated with high vegetable intake (dark leafy vegetables and gourds/shoots/root vegetables).

The present study found that dark leafy vegetables were positively correlated with hepcidin levels. To put our results into perspective, we observed that one frequency increase in dark leafy vegetable consumption increased serum hepcidin levels by 0.013 and 0.012 ng/mL in pregnant women with IDA and ID, respectively (all *p* < 0.005). Vegetables are polyphenol-rich foods. The literature shows that polyphenols may either up- or downregulate hepcidin expression. Zen and colleagues found that genistein stimulates hepcidin expression via enhanced signal transduction and activator of transcription 3 (STAT3) in HepG2 cells [34]. An animal study revealed that quercetin inhibited intestinal non-heme iron absorption via decreasing duodenal divalent metal transporter 1 (DMT1), duodenal cytochrome B (Dcytb), and ferroportin but increasing hepcidin expression in the liver and spleen [23]. In contrast, Mu et al. found that myricetin potently inhibited hepcidin expression, and this effect was mediated by BMP/SMAD signaling [22]. An in vivo study showed that epigallocatechin-3-gallate (EGCG) inhibited interleukin (IL)-6-induced hepcidin [35]. Overall, polyphenols may influence the relationship between vegetable consumption and the risk of gestational ID/IDA.

Vitamin C may also play a role in vegetable consumption and the IDA risk. The risk of IDA was 2.47- and 1.92-fold higher among pregnant women who consumed the lowest vitamin C (T1) but highest levels of dark leafy vegetables (T3: 49 times/month) and gourds/shoots/root vegetables (T3: 55 times/month), respectively, compared to those who consumed the lowest levels. The risk decreased with increasing vitamin C consumption (T3: 194 mg/day) for women with the highest intake of dark leafy vegetables (T3) and gourds/shoots/root vegetables (T3). Ghatpande el at. found that low consumption of vitamin C-rich fruits was associated with lower levels of circulating iron [16]. Vegetables contain a high amount of dietary iron. The bioavailability of plant-based iron depends upon the content of promoters (vitamin C) and inhibitors (phytates, tannins, and polyphenols) [16]. Plant iron is the non-heme iron, ferric iron (Fe^3+^), which needs to be reduced to the soluble ferrous iron (Fe^2+^) for absorption. This is achieved by the combined actions of Dcytb, a ferrireductase that is present on apical membranes of duodenal enterocytes and dietary reducing agents, such as vitamin C [36]. Importantly, vitamin C may directly enhance iron bioavailability by inhibiting hepcidin mRNA expression [37].

Another interesting finding was the correlations among carbohydrate-rich foods, hepcidin, and gestational ID/IDA. The present study found that one frequency increase in Chinese dim sum and related foods was associated with a 0.037 increase in hepcidin levels, but one frequency increase in noodles and related products was correlated with a 0.022 decrease in hepcidin (all *p* < 0.005). The adjusted logistic regression showed that total carbohydrate intake [OR: 1.003 (1.000~1.006), *p* = 0.036] predicted the ID risk, and the intake frequency of rice/rice porridge [OR: 1.007 (1.00~1.014), *p* = 0.041] predicted gestational IDA. According to the food classification, Chinese dim sum and related foods and rice/rice porridge are classified as foods with a high glycemic index (GI), but noodles and related products are classified as foods with a low GI [38,39]. Dickinson et al. found that among young lean healthy subjects, consumption of high-GI foods was associated with increased oxidative stress and both acute and chronic low-grade inflammation [40]. There is general agreement in the literature that the presence of oxidative stress and inflammation stimulates hepcidin over-synthesis, which in turn, leads to functional ID and IDA [6.7]. Our previous study showed that an altered carbohydrate/fat ratio predicted functional IDA among overweight and obese women [41]. Animal studies showed that a high-fructose diet induces a systemic iron deficiency and hepatic iron overload, possibly by inflammation-mediated hepcidin-ferroportin alterations [42]. Since lowering maternal hepcidin is essential for transporting bioavailable iron to the placenta [9], future studies need to clarify the underlying mechanisms between the type of carbohydrate foods consumed and hepcidin levels among pregnant women.

Several limitations of this study should be recognized when interpreting the results. Because NAHSIT-Pregnant is a cross-sectional survey, the present study could not assess temporality or causality in any of the associations we found. Our analysis included only one 24-h set of recall data, which might not accurately reflect the usual intake of each individual but does provide a valid means for estimating the intake of the group. Our findings also largely rely on self-reported dietary data, which may be subject to recall bias or systematic underreporting of energy intake, particularly in those who are overweight or obese in late pregnancy [43]. In addition, we were unable to account for the effects of pregnancy-related nausea and vomiting that typically occur in the first trimester on dietary intake; thus, it is possible that self-reported dietary intake was overestimated in this subgroup. Failure to account for total energy intake can obscure associations between nutrient intake levels and disease risk or may even reverse the direction of the association. Thus, the current study adjusted for the total energy intake as a way to control confounding dietary factors [33]. Over-reporting of food intake in FFQs was observed in pregnant women [44]. However, the FFQ is considered to be one of the most appropriate dietary methods used in large-scale epidemiological studies, and it provides useful information on dietary patterns [45]. Despite these limitations, the present study had several strengths. The main strength of the current study was the use of a nationwide representative sample of pregnant women in Taiwan, which also provided a sufficiently large sample population. This suggests that our estimates may have good precision and/or reliability. The present study also included various parameters, which allowed us to adjust for several potential confounding variables that are known to affect gestational IDA, such as age, pre-pregnancy BMI, parity, total supplement use (%), and trimester.

## 5. Conclusions

Intake frequency of carbohydrate-rich foods and vegetables may affect the gestational iron status through influencing hepcidin levels. Specifically, the increased consumption of total carbohydrate intake, especially refined carbohydrates may increase the risk of gestational ID and IDA. By contrast, complex carbohydrates foods (e.g., breakfast cereals, oats, and related products) may protect against gestational ID and IDA through downregulating hepcidin levels to enhance iron delivery to the fetus. A high intake of vegetables may also increase the risk, but this effect can be suppressed by the addition of vitamin C in the diet. Additional studies are needed to confirm the food-hepcidin-ID/IDA relationship among pregnant women living in countries with different cultural contexts and food preferences. Overall, dietary strategies that incorporate hepcidin-modulating foods may help prevent gestational ID or IDA.

## Figures and Tables

**Figure 1 nutrients-13-03501-f001:**
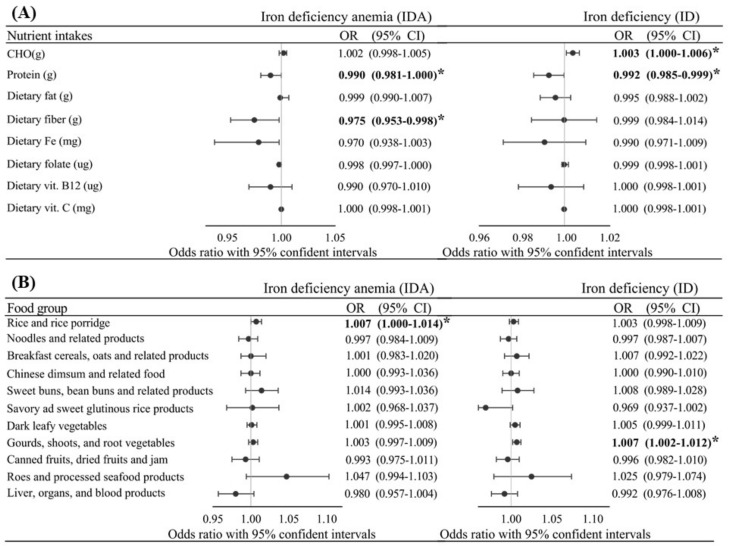
Adjusted odds ratio (OR) and 95% confidence intervals (CI) of gestational iron deficiency (ID) and ID anemia (IDA) with various intake levels of nutrients (**A**) and foods (**B**) among all pregnant women after adjusting for age, pre-pregnancy body-mass index, region, parity, total supplement use (%), and trimester. *p* value/*p* for trend as ***** *p* ≤ 0.05.

**Figure 2 nutrients-13-03501-f002:**
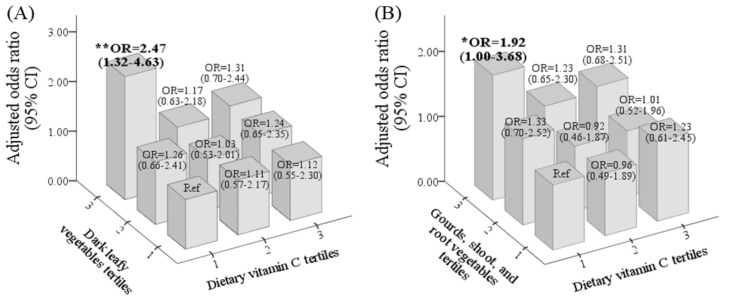
Odd ratio (OR) and 95% confidence interval (CI) of iron-deficiency anemia (IDA) for the interactive effect of vitamin C and dark leafy vegetables (**A**) and gourds/shoots/root vegetables (**B**) after adjusting for age, pre-pregnancy body-mass index, region, parity, total supplement use (%), and trimester. *p* value/*p* for trend as ***** *p* ≤ 0.05, ****** *p* ≤ 0.01.

**Table 1 nutrients-13-03501-t001:** Maternal baseline characteristics according to quartiles of serum hepcidin levels (*N* = 1430).

Variable	Serum Hepcidin Level, Quartiles				*p* for Trend ^a^
	Q_1_ (*n* = 355)	Q_2_ (*n* = 360)	Q_3_ (*n* = 358)	Q_4_ (*n* = 357)	
Basic characteristic					
Age (years)	31.7 ± 5.0	32.9 ± 4.6	32.8 ± 4.6	32.9 ± 4.6	0.002
Pre-pregnancy body-mass index (kg/m^2^)	22.1 ± 3.9	22.8 ± 4.0	22.7 ± 4.1	23.0 ± 4.1	0.006
Region					<0.001
Northern (*n*, %)	95 (26.8)	115 (31.9)	132 (36.9)	136 (38.1)	
Central (*n*, %)	79 (22.3)	95 (26.4)	88 (24.6)	67 (18.8)	
Southern (*n*, %)	67 (18.9)	65 (18.1)	80 (22.3)	74 (20.7)	
Eastern (*n*, %)	114 (32.1)	85 (23.6)	58 (16.2)	80 (22.4)	
Trimester					<0.001
First trimester (*n*, %)	43 (12.1)	36 (10.0)	97 (27.1)	178 (49.9)	
Second trimester (*n*, %)	112 (31.5)	84 (23.3)	145 (40.5)	122 (34.2)	
Third trimester (*n*, %)	200 (56.3)	240 (66.7)	116 (32.4)	57 (16.0)	
Parity					<0.001
Primiparous (*n*, %)	158 (44.6)	191 (53.2)	215 (60.2)	212 (59.7)	
Reported use of prenatal dietary supplements					
Total supplement use (*n*, %)	259 (73.4)	291 (82.7)	320 (90.4)	321 (90.9)	<0.001
Multivitamin-mineral (*n*, %)	199 (56.5)	220 (63.2)	246 (69.5)	216 (61.4)	0.071
Vitamin B (*n*, %)	47 (13.5)	61 (17.4)	74 (21.1)	73 (20.9)	0.005
Folate (*n*, %)	138 (39.2)	145 (41.4)	165 (47.3)	195 (55.7)	<0.001
Iron (*n*, %)	33 (9.5)	44 (12.6)	33 (9.3)	37 (10.6)	0.988
Blood biochemistry					
Serum hepcidin (ng/mL)	0.3 ± 0.3	2.4 ± 1.4	19.8 ± 10.6	73.2 ± 24.6	<0.001
TS (%)	10.1 ± 7.5	12.8 ± 7.9	20.8 ± 9.3	22.2 ± 9.1	<0.001
Hb (g/dL)	11.2 ± 1.8	11.5 ± 1.7	12.2 ± 2.0	12.1 ± 2.0	<0.001
Serum ferritin (ng/mL)	8.0 ± 6.0	10.3 ± 4.4	23.8 ± 14.6	50.0 ± 33.7	<0.001
Iron abnormality					
Anemia (*n*, %)	142 (40.0)	104 (28.9)	51 (14.2)	57 (16.0)	<0.001
Non-IDA anemia (*n*, %)	10 (2.8)	12 (3.3)	39 (10.9)	57 (16.0)	<0.001
IDA (*n*, %)	132 (37.2)	92 (25.6)	12 (3.4)	0 (0.0)	<0.001
Iron deficiency (*n*, %)	293 (82.5)	263 (73.1)	49 (13.7)	1 (0.3)	<0.001

^a^*p* for trend was performed by a general linear model for categorical variables and a one-way ANOVA for continuous variables. Continuous data are presented as the mean ± standard deviation (SD); categorical data are presented as the number (percentage of the same group). Hepcidin quartiles: quartile 1 (Q_1_) ≤0.80 ng/mL; quartile 2 (Q_2_) >0.81–≤5.66 ng/mL; quartile 3 (Q_3_) >5.67–≤40.05 ng/mL; quartile 4 (Q_4_) ≥40.00 ng/mL. Abbreviation: TS, transferrin saturation; Hb, hemoglobin; IDA, iron-deficiency anemia. Definitions, anemia: first trimester Hb <11 g/dL, second trimester Hb <10.5 g/dL, third trimester Hb <11 g/dL; IDA: serum ferritin ˂15 ng/mL and transferrin saturation <16%, and first trimester Hb <11 g/dL, second trimester Hb <10.5 g/dL, third trimester Hb <11 g/dL, ID: serum ferritin <15 ng/mL and transferrin saturation <16%.

**Table 2 nutrients-13-03501-t002:** Nutrient and food intake frequencies among pregnant women according to quartile levels of serum hepcidin (*N* = 1430).

Variable	Hepcidin-Level, Quartiles				*p* for Trend ^a^
	Quartile 1 (*n* = 355)	Quartile 2 (*n* = 360)	Quartile 3 (*n* = 358)	Quartile 4 (*n* = 357)	
Nutrient intake ^b^					
Carbohydrates (g)	14.3 ± 48.7	13.6 ± 45.3	11.3 ± 43.3	9.7 ± 45.0	0.141
Protein (g)	5.0 ± 17.7	6.5 ± 18.2	7.8 ± 16.9	6.8 ± 16.0	0.099
Fat (g)	−8.0 ± 19.8	−7.9 ± 17.4	−7.6 ± 18.1	−6.4 ± 18.9	0.255
Dietary fiber (g)	1.5 ± 8.6	2.2 ± 8.2	2.0 ± 7.3	3.0 ± 9.4	0.04
Dietary Fe (mg)	2.0 ± 4.9	2.8 ± 6.8	2.2 ± 4.7	3.0 ± 8.6	0.108
Dietary B9 (µg)	74.2 ± 83.0	76.8 ± 83.1	76.2 ± 84.0	78.2 ± 108.3	0.594
Dietary B12 (µg)	2.9 ± 10.3	2.5 ± 7.3	2.2 ± 5.9	2.9 ± 9.6	0.941
Dietary vitamin C (mg)	67.7 ± 100.3	66.8 ± 108.0	80.8 ± 119.0	86.0 ± 145.2	0.014
Food group ^b^ (frequency: times/month)					
Rice and rice porridge	38.6 ± 23.2	36.4 ± 21.7	36.0 ± 23.0	35.8 ± 21.6	0.096
Noodles and related products	15.5 ± 13.4	15.0 ± 12.0	16.0 ± 12.2	16.1 ± 12.9	0.303
Breakfast cereals, oats, and related products	2.9 ± 9.3	1.7 ± 7.0	2.0 ± 8.3	1.4 ± 8.5	0.029
Chinese dim sum and related foods	9.0 ± 11.6	9.6 ± 11.8	10.0 ± 12.3	10.5 ± 13.4	0.144
Sweet buns, bean buns, and related products	3.2 ± 6.2	2.8 ± 8.0	1.6 ± 4.6	2.1 ± 5.1	0.002
Savory and sweet glutinous rice products	0.9 ± 3.3	0.9 ± 3.1	0.67 ± 2.8	1.3 ± 6.8	0.512
Light-colored vegetables	23.4 ± 21.3	24.1 ± 22.0	21.3 ± 18.0	21.2 ± 19.2	0.05
Dark leafy vegetables	26.4 ± 20.8	29.8 ± 23.4	26.6 ± 21.0	24.9 ± 18.7	0.121
Gourds, shoots, and root vegetables	29.3 ± 26.2	30.1 ± 27.5	26.0 ± 22.9	25.4 ± 22.8	0.007
Legumes	4.0 ± 7.9	3.6 ± 7.5	3.1 ± 5.5	2.7 ± 4.8	0.007
Total vegetables	47.2 ± 27.6	48.9 ± 29.8	44.8 ± 28.0	45.2 ± 27.6	0.135
Fresh fruits	32.6 ± 20.9	32.2 ± 21.3	29.4 ± 18.5	30.1 ± 20.0	0.029
100% pure fruit juice or fruit/vegetable juice	5.4 ± 8.6	5.4 ± 9.5	4.8 ± 6.6	7.3 ± 12.9	0.033
Canned fruits, dried fruits, and jam	2.3 ± 9.6	2.1 ± 7.8	1.8 ± 7.9	4.4 ± 14.5	0.016
Fish, shellfish, and seafood	20.9 ± 20.1	21.5 ± 19.6	17.7 ± 16.4	19.1 ± 18.0	0.043
Roe and processed seafood products	0.6 ± 2.3	0.5 ± 2.76	0.4 ± 2.0	0.7 ± 2.7	0.845
Poultry meat	17.1 ± 15.4	17.4 ± 15.6	16.7 ± 14.4	15.6 ± 13.1	0.14
Liver, organs, and blood products	2.8 ± 8.5	3.0 ± 7.2	2.6 ± 5.9	3.9 ± 9.6	0.123

^a^ *p* for trend was assessed by a one-way ANOVA. All data are presented as the mean ± standard deviation (SD). ^b^ Nutrient intake and food group were adjusted for total energy using residual methods. Hepcidin quartiles: quartile 1 (Q_1_) ≤0.80 ng/mL; quartile 2 (Q_2_) >0.81–≤5.66 ng/mL; quartile 3 (Q_3_) >5.67–≤40.05 ng/mL; quartile 4 (Q_4_) ≥40.00 ng/mL.

**Table 3 nutrients-13-03501-t003:** Adjusted beta coefficient (ß) and 95% confidence interval (CI) of gestational serum hepcidin levels for nutrients and food groups among pregnant women.

	All Pregnant Women (*N* = 1430)		Non-Iron Deficiency/Iron-Deficiency Anemia (Non-ID/IDA) Pregnant Women (*N* = 824)		Iron Deficiency (ID) Pregnant Women (*N* = 606)		Iron Deficiency Anemia (IDA) Pregnant Women (*N* = 236)	
	ß (95% CI)	*p* Value ^a^	ß (95% CI)	*p* Value ^a^	ß (95% CI)	*p* Value ^a^	ß (95% CI)	*p* Value ^a^
Nutrient intake ^b^								
Carbohydrates (g)	−0.027 (−0.061–0.007)	0.122	−0.016 (−0.068–0.036)	0.554	0.000 (−0.006–0.005)	0.891	−0.001 (−0.004–0.002)	0.531
Protein (g)	0.023 (−0.067–0.113)	0.617	−0.051 (−0.188–0.086)	0.465	−0.004 (−0.019–0.011)	0.574	−0.004 (−0.014–0.005)	0.382
Fat (g)	0.071 (−0.014–0.155)	0.100	0.069 (−0.059–0.196)	0.291	0.003 (−0.011–0.018)	0.652	−0.002 (−0.010–0.007)	0.711
Dietary fiber (g)	0.121 (−0.063–0.306)	0.198	0.184 (−0.093–0.460)	0.192	−0.010 (−0.042–0.021)	0.525	0.001 (−0.038–0.040)	0.969
Dietary Fe (mg)	0.203 (−0.032–0.439)	0.091	0.209 (−0.104–0.522)	0.191	−0.029 (−0.080–0.021)	0.255	−0.026 (−0.080–0.027)	0.333
Dietary folate (µg)	0.008 (−0.009–0.025)	0.338	0.006 (−0.018–0.031)	0.602	−0.002 (−0.005–−0.001)	0.243	−0.002 (−0.005–0.001)	0.201
Dietary vitamin B12 (µg)	0.117 (−0.002–0.355)	0.052	0.268 (−0.014–0.550)	0.062	−0.011 (−0.039–0.018)	0.451	−0.017 (−0.047–0.012)	0.254
Dietary vitamin C (mg)	0.014 (0.001–0.026)	0.041	0.014 (−0.004–0.031)	0.136	−0.000 (−0.003–0.002)	0.942	0.001 (−0.002–0.004)	0.560
Food group ^b^ (frequency: times/month)								
Rice and rice porridge	−0.035 (−0.103–0.034)	0.318	−0.023 (−0.126–0.080)	0.658	−0.009 (−0.021–0.002)	0.123	−0.006 (−0.016–0.004)	0.233
Noodles and related products	0.071 (−0.051–0.192)	0.255	0.070 (−0.113–−0.253)	0.453	−0.008 (−0.029–0.012)	0.421	−0.022 (−0.043–−0.001)	0.038
Breakfast cereals, oats, and related products	−0.187 (−0.370–−0.003)	0.046	−0.295 (−0.567–−0.023)	0.034	0.027 (−0.005–0.058)	0.097	0.005 (−0.026–0.037)	0.737
Chinese dim sum and related foods	0.033 (−0.090–0.156)	0.603	0.063 (−0.123–0.249)	0.505	0.007 (−0.014–0.028)	0.506	0.037 (0.015–0.058)	0.001
Sweet buns, bean buns, and related products	0.097 (−0.347–0.152)	0.445	−0.160 (−0.614–0.294)	0.489	−0.019 (−0.055–0.016)	0.286	−0.001 (−0.027–0.026)	0.969
Savory and sweet glutinous rice products	0.677 (0.330–1.024)	<0.001	0.662 (0.218–1.107)	0.004	−0.007 (−0.091–0.076)	0.862	0.017 (−0.064–0.097)	0.683
Light–colored vegetables	−0.075 (−0.150–0.001)	0.052	−0.079 (−0.201–0.042)	0.201	0.003 (−0.009–0.015)	0.602	0.003 (−0.011–0.017)	0.693
Dark leafy vegetables	−0.061 (−0.133–0.012)	0.101	−0.079 (−0.195–0.037)	0.181	0.012 (0.000–0.023)	0.006	0.013 (0.001–0.025)	0.04
Gourds, shoots, and root vegetables	−0.071 (−0.133–−0.010)	0.022	−0.066 (−0.165–0.033)	0.190	0.004 (−0.006–0.014)	0.403	−0.009 (−0.018–0.001)	0.076
Legumes	−0.129 (−0.362–0.104)	0.276	−0.045 (−0.456–0.366)	0.831	−0.015 (−0.049–0.019)	0.375	−0.009 (−0.052–0.033)	0.666
Total vegetables	−0.024 −0.079–0.030)	0.376	−0.022 (−0.105–0.060)	0.595	0.007 (−0.002–0.016)	0.105	0.003 (−0.006–0.012)	0.541
Fresh fruits	−0.046 (−0.122–0.030)	0.237	−0.022 (−0.142–0.098)	0.716	0.008 (0.005–0.020)	0.220	0.003 (−0.009–0.015)	0.637
100% pure fruit juice or fruit/vegetable juice	0.140 (−0.017–0.298)	0.08	0.173 (−0.057–0.404)	0.140	0.018 (−0.010–0.046)	0.198	−0.010 (−0.035–0.014)	0.392
Canned fruits, dried fruits, and jam	0.157 (0.009–0.304)	0.037	0.176 (−0.018–0.370)	0.076	−0.014 (−0.046–0.019)	0.415	−0.005 (−0.042–0.032)	0.8
Fish, shellfish, and seafood	−0.014 (−0.097–0.068)	0.733	−0.001 (−0.132–0.130)	0.991	−0.014 (−0.046–0.019)	0.415	−0.002 (−0.015–0.011)	0.764
Roe and processed seafood products	0.434 (−0.153–1.022)	0.147	1.223 (0.321–2.124)	0.008	−0.021 (−0.118–0.077)	0.678	−0.015 (−0.099–0.070)	0.731
Poultry meat	−0.076 (−0.180–0.028)	0.151	−0.083 (−0.249–0.082)	0.323	−0.010 (−0.027–0.006)	0.226	−0.007 (−0.022–0.008)	0.378
Liver, organs, and blood products	0.332 (0.139–0.526)	0.001	0.489 (0.189–0.790)	0.001	0.001 (−0.033–0.030)	0.927	0.012 (−0.029–0.053)	0.558

^a^ Adjusted for age, pre-pregnancy body-mass index, region, parity, total supplement use (%), and trimester. ^b^ Nutrient intake and food group were adjusted for total energy using residual method.

## Data Availability

The datasets used and/or analyzed during the current study are available from the corresponding author on reasonable request.

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
