# Peer review of "Associations of Food and Nutrient Intake with Serum Hepcidin and the Risk of Gestational Iron-Deficiency Anemia among Pregnant Women: A Population-Based Study"

_nutrients, 2021, doi:10.3390/nu13103501_

Round 1
Reviewer 1 Report
This is an interesting manuscript presenting data not otherwise available.
There are some opportunities for improvement in the manuscript.
1. In Table 2, it should be made clear that the category "Energy adjusted of food group" (I think the "of" is redundant and can be removed) is expressed in portions per month. On a related issue, the definitions of the various quartiles in Table 2 should not be repeated in the text description of the results since they are stated in the Table
2. In the introduction, outlines 59 – 62 the prevalence of gestational iron deficiency in North America is reported to be very low in the first trimester. Recent studies from North America have suggested the incidence is significantly higher [Auerbach 2019; Teichman 2021). These more recent data should be commented upon, and if the authors wish, they should say why it differs from what is reported in the reference they cite.
3. One of the key pathophysiology issues in this paper is whether the effects observed with hepcidin are a primary effect of some dietary factor or are a secondary effect of changes in iron absorption. The authors address this with reasonable explicitness for vitamin C and for quercetin. However, they have the opportunity to address it more directly by controlling for serum iron concentration, which is data that was collected. They do control for iron "levels" to some degree by comparing iron deficient values to non-iron deficient values, but a specific assessment of potential contribution of serum iron concentration in the overall population as well as in the iron deficient and iron replete subsets is an opportunity for increased understanding that should not be missed.
Author Response
Reviewer 1
This is an interesting manuscript presenting data not otherwise available.
There are some opportunities for improvement in the manuscript.
1. In Table 2, it should be made clear that the category "Energy adjusted of food group" (I think the "of" is redundant and can be removed) is expressed in portions per month. On a related issue, the definitions of the various quartiles in Table 2 should not be repeated in the text description of the results since they are stated in the Table
Response 1:
Thank you for your comments and suggestions. We modified table 2 and the footnote accordingly as below. We also deleted the repeated sentences in the text (The mean serum hepcidin levels for Q1, Q2, Q3, and Q4 were 0.3±0.3, 2.4±1.4, 19.8±10.6, and 73.2±24.6 ng/mL, respectively (p for trend <0.001) (Table 1) (Page 5, lines 219-220 of original manuscript).
- Original Table 2
Table 2. Nutrient and food intake frequencies among pregnant women according to quartile levels of serum hepcidin (N=1430)
Variable |
Hepcidin-level, Quartiles |
p for trend a |
||||||||||
Quartile 1 (n=355) |
Quartile 2 (n=360) |
Quartile 3 (n=358) |
Quartile 4 (n=357) |
|||||||||
Energy-adjusted of nutrient intake (residual methods) |
||||||||||||
Carbohydrates (g) |
14.3 ± 48.7 |
13.6 ± 45.3 |
11.3 ± 43.3 |
9.7 ± 45.0 |
0.141 |
|
||||||
Protein (g) |
5.0 ± 17.7 |
6.5 ± 18.2 |
7.8 ± 16.9 |
6.8 ± 16.0 |
0.099 |
|
||||||
Fat (g) |
-8.0 ± 19.8 |
-7.9 ± 17.4 |
-7.6 ± 18.1 |
-6.4 ± 18.9 |
0.255 |
|
||||||
Dietary fiber (g) |
1.5 ± 8.6 |
2.2 ± 8.2 |
2.0 ± 7.3 |
3.0 ± 9.4 |
0.04 |
|
||||||
Dietary Fe (mg) |
2.0 ± 4.9 |
2.8 ± 6.8 |
2.2 ± 4.7 |
3.0 ± 8.6 |
0.108 |
|
||||||
Dietary B9 (µg) |
74.2 ± 83.0 |
76.8 ± 83.1 |
76.2 ± 84.0 |
78.2 ± 108.3 |
0.594 |
|
||||||
Dietary B12 (µg) |
2.9 ± 10.3 |
2.5 ± 7.3 |
2.2 ± 5.9 |
2.9 ± 9.6 |
0.941 |
|
||||||
Dietary vitamin C (mg) |
67.7 ± 100.3 |
66.8 ± 108.0 |
80.8 ± 119.0 |
86.0 ± 145.2 |
0.014 |
|
||||||
Energy-adjusted of food group (residual methods) |
||||||||||||
|
Rice and rice porridge |
38.6 ± 23.2 |
36.4 ± 21.7 |
36.0 ± 23.0 |
35.8 ± 21.6 |
0.096 |
||||||
|
Noodles and related products |
15.5 ± 13.4 |
15.0 ± 12.0 |
16.0 ± 12.2 |
16.1 ± 12.9 |
0.303 |
||||||
|
Breakfast cereals, oats, and related products |
2.9 ± 9.3 |
1.7 ± 7.0 |
2.0 ± 8.3 |
1.4 ± 8.5 |
0.029 |
||||||
|
……………………. |
|
||||||||||
a p for trend was assessed by a one-way ANOVA. All data are presented as the mean ± standard deviation (SD). Hepcidin quartiles: quartile 1 (Q1) ≤ 0.80 ng/mL; quartile 2 (Q2) >0.81~≤5.66 ng/mL; quartile 3 (Q3) >5.67~≤40.05 ng/mL; quartile 4 (Q4) ≥40.00 ng/mL.
- Revision Table 2 (Page 7, Lines 260-262)
Table 2. Nutrient and food intake frequencies among pregnant women according to quartile levels of serum hepcidin (N=1430).
Variable |
Hepcidin-level, Quartiles |
p for trend a |
||||
Quartile 1 (n=355) |
Quartile 2 (n=360) |
Quartile 3 (n=358) |
Quartile 4 (n=357) |
|||
Nutrient intakeb |
||||||
Carbohydrates (g) |
14.3 ± 48.7 |
13.6 ± 45.3 |
11.3 ± 43.3 |
9.7 ± 45.0 |
0.141 |
|
Protein (g) |
5.0 ± 17.7 |
6.5 ± 18.2 |
7.8 ± 16.9 |
6.8 ± 16.0 |
0.099 |
|
Fat (g) |
-8.0 ± 19.8 |
-7.9 ± 17.4 |
-7.6 ± 18.1 |
-6.4 ± 18.9 |
0.255 |
|
Dietary fiber (g) |
1.5 ± 8.6 |
2.2 ± 8.2 |
2.0 ± 7.3 |
3.0 ± 9.4 |
0.04 |
|
Dietary Fe (mg) |
2.0 ± 4.9 |
2.8 ± 6.8 |
2.2 ± 4.7 |
3.0 ± 8.6 |
0.108 |
|
Dietary B9 (µg) |
74.2 ± 83.0 |
76.8 ± 83.1 |
76.2 ± 84.0 |
78.2 ± 108.3 |
0.594 |
|
Dietary B12 (µg) |
2.9 ± 10.3 |
2.5 ± 7.3 |
2.2 ± 5.9 |
2.9 ± 9.6 |
0.941 |
|
Dietary vitamin C (mg) |
67.7 ± 100.3 |
66.8 ± 108.0 |
80.8 ± 119.0 |
86.0 ± 145.2 |
0.014 |
|
Food groupb (frequency: times /month) |
||||||
|
Rice and rice porridge |
38.6 ± 23.2 |
36.4 ± 21.7 |
36.0 ± 23.0 |
35.8 ± 21.6 |
0.096 |
|
Noodles and related products |
15.5 ± 13.4 |
15.0 ± 12.0 |
16.0 ± 12.2 |
16.1 ± 12.9 |
0.303 |
|
Breakfast cereals, oats, and related products |
2.9 ± 9.3 |
1.7 ± 7.0 |
2.0 ± 8.3 |
1.4 ± 8.5 |
0.029 |
|
Chinese dim sum and related foods |
9.0 ± 11.6 |
9.6 ± 11.8 |
10.0 ± 12.3 |
10.5 ± 13.4 |
0.144 |
|
Sweet buns, bean buns, and related products |
3.2 ± 6.2 |
2.8 ± 8.0 |
1.6 ± 4.6 |
2.1 ± 5.1 |
0.002 |
|
…………………………. |
|
|
|
|
|
a p for trend was assessed by a one-way ANOVA. All data are presented as the mean ± standard deviation (SD). b Nutrient intake and food group were adjusted for total energy using residual methods. Hepcidin quartiles: quartile 1 (Q1) ≤ 0.80 ng/mL; quartile 2 (Q2) >0.81~≤5.66 ng/mL; quartile 3 (Q3) >5.67~≤40.05 ng/mL; quartile 4 (Q4) ≥40.00 ng/mL.
- Results (Page 5, Lines 215-220)
“Table 1 shows baseline characteristics of study participants according to serum hepcidin levels. Compared to those with the lowest (Q1) serum hepcidin levels, pregnant women with the highest serum hepcidin (Q4) were older, heavier, had the highest proportions of living in the northern area of Taiwan (38.1%), a primiparous rate (59.7%), in trimester 1 (49.9%), and used total prenatal supplements (90.9%) and folate supplements (55.7%) (all p for trend <0.01) (Table 1).”
- In the introduction, outlines 59 – 62 the prevalence of gestational iron deficiency in North America is reported to be very low in the first trimester. Recent studies from North America have suggested the incidence is significantly higher [Auerbach 2019; Teichman 2021). These more recent data should be commented upon, and if the authors wish, they should say why it differs from what is reported in the reference they cite.
Response 2:
We totally agree with your comment. We modified the introduction as below:
Introduction (Page 2, Lines 59-64)
“In North America, the prevalence of gestational ID is very low (7.3%) in the first trimester but significantly increases in the second (23.7%) and third trimesters (39.2%)[3]. This corresponds to a progressive rise in gestational anemia in the first, second, and third trimesters: 2.7%, 2.2%, and 10.8%, respectively[3]. However, a recent survey shows an increasing trend of gestational ID (23.8%) in the first trimester in 2021 [4]. “
- One of the key pathophysiology issues in this paper is whether the effects observed with hepcidin are a primary effect of some dietary factor or are a secondary effect of changes in iron absorption. The authors address this with reasonable explicitness for vitamin C and for quercetin. However, they have the opportunity to address it more directly by controlling for serum iron concentration, which is data that was collected. They do control for iron "levels" to some degree by comparing iron deficient values to non-iron deficient values, but a specific assessment of potential contribution of serum iron concentration in the overall population as well as in the iron deficient and iron replete subsets is an opportunity for increased understanding that should not be missed.
Response 3:
Thank you for your comment. We do not fully understand your comment but from the literature knowledge, we know that blood iron biochemistry (e.g., TS%) indices have a stronger correlation with hepcidin than food groups. After we further adjusted TS% (Model 2) as below, the significant correlation between food group (e.g., Gourds, shoots, and root vegetables), nutrient (e.g., vitamin C), and hepcidin disappear. This suggests that hepcidin has a stronger correlation with iron biomarkers than food group and nutrients. However, some of the results remain similar (e.g., Dark leafy vegetables, Chinese dim sum, and related food), suggesting food group and nutrients independent effects on hepcidin.
|
Model 1 |
Model 2 |
|||||||
ß (95% CI) |
p-value |
ß (95% CI) |
p-value |
||||||
All pregnant women |
|
|
|
|
|||||
Dietary vit C (mg) |
0.014 (0.001-0.026) |
0.041 |
0.011 (-0.001-0.023) |
0.076 |
|||||
Breakfast cereals, oats and related products |
-0.187 (-0.370--0.003) |
0.046 |
-0.156 (-0.330-0.018) |
0.079 |
|||||
Savory and sweet glutinous rice product |
0.677 (0.330-1.024) |
<0.001 |
0.591 (0.261-0.920) |
<0.001 |
|||||
Gourds, shoots and root vegetables |
-0.071 (-0.133--0.010) |
0.022 |
-0.057 (-0.115-0.002) |
0.056 |
|||||
Canned fruits, dried fruits and jam |
0.157 (0.009-0.304) |
0.037 |
0.116 (-0.024-0.256) |
0.104 |
|||||
Iron Deficiency (ID) pregnant women |
|||||||||
Dark leafy vegetable vegetables |
0.012 (-0.000-0.023) |
0.006 |
0.012 (-0.000-0.023) |
0.043 |
|
||||
Iron Deficiency Anemia (IDA) pregnant women |
|||||||||
Noodles and related products
|
-0.022 (-0.043--0.001) |
0.038 |
-0.016 (-0.035-0.003) |
0.1 |
|||||
Dark leafy vegetable
|
0.013 (0.001-0.025) |
0.04 |
0.010 (-0.001-0.022) |
0.075 |
|||||
Chinese dimsum and related food |
0.037 (-0.015-0.058) |
0.001 |
0.032 (0.012-0.052) |
0.002 |
|||||
Model 1= Adjusted for age, pre-pregnancy body-mass index, region, parity, total supplement use (%), and trimester.
Model 2 =Adjusted for age, pre-pregnancy body-mass index, region, parity, total supplement use (%), trimester, and TS (%).

Reviewer 2 Report
Interesting study and well conducted
Author Response
Reviewer 2
Interesting study and well conducted
Response: Thank you for your positive response to our work.

Reviewer 3 Report
Overall, the research has been well conducted and the paper is informative and makes a positive contribution to knowledge in the domain of nutrition and health.
There are only a few minor points.
Given that the paper is on hepcidin, it needs to be described in the first part of the abstract. Something as simple as, Hepcidin is a regulator of iron metabolism. Although the abstract is long, the implication of the findings to the domain of nutrition and health still needs to be included. Some of the Beta values could be summarised.
The conclusion is also very short and needs extended. The food survey of Taiwanese women has a particular cultural context and so what is the message from this research to the wider international nutrition and health audience and to international medical practitioners? The implications of the research needs to be extended.
There continues to be an ongoing debate about iron supplementation and pregnant women, but this issue was not raised in the research paper. Given the research findings on hepcidin, do the researchers have a comment on iron supplementation and pregnant women? See the research of Bah et al. on this topic: Bah, A., Wegmuller, R., Cerami, C. et al. A double blind randomised controlled trial comparing standard dose of iron supplementation for pregnant women with two screen-and-treat approaches using hepcidin as a biomarker for ready and safe to receive iron. BMC Pregnancy Childbirth 16, 157 (2016). https://doi.org/10.1186/s12884-016-0934-8
Again, a well-developed and written paper and I congratulate the researchers involved.
Author Response
Reviewer 3
Overall, the research has been well conducted and the paper is informative and makes a positive contribution to knowledge in the domain of nutrition and health.
Response: Thank you for your positive response to our work.
There are only a few minor points.
- Given that the paper is on hepcidin, it needs to be described in the first part of the abstract. Something as simple as, Hepcidin is a regulator of iron metabolism. Although the abstract is long, the implication of the findings to the domain of nutrition and health still needs to be included. Some of the Beta values could be summarised.
Response 1:
Thank you for your comments. We added new sentences in the abstract accordingly as below:
Abstract (Page 1, Line 29)
“Hepcidin is a regulator of iron metabolism”
2.The conclusion is also very short and needs extended. The food survey of Taiwanese women has a particular cultural context and so what is the message from this research to the wider international nutrition and health audience and to international medical practitioners? The implications of the research needs to be extended.
Response 2:
We totally agree with your comments as food survey data is regional with cultural context. However, the message from this research can be used for international audience. We have extent the conclusion and provide the implications of the research needs as bellows:
- Conclusion (Page 12, Lines 398-408)
“Intake frequency of carbohydrates and vegetables may affect the gestational iron status through influencing hepcidin levels. Specifically, the increased consumption of total carbohydrate intake, especially refined carbohydrates may increase the risk of gestational ID and IDA. By contrast, complex carbohydrates foods (e.g., breakfast cereals, oats, and related products) may protect against gestational ID and IDA through downregulating hepcidin levels to enhance iron delivery to the fetus. High intakes of vegetable may also increase the risk but this effect can be suppressed by the addition of vitamin C in the diet. Additional studies are needed to confirm the food-hepcidin-ID/IDA relationship among pregnant women living in countries with different cultural context and food preferences. Overall, dietary strategies that incorporate hepcidin-modulating foods may help prevent gestational ID or IDA.”
3.There continues to be an ongoing debate about iron supplementation and pregnant women, but this issue was not raised in the research paper. Given the research findings on hepcidin, do the researchers have a comment on iron supplementation and pregnant women? See the research of Bah et al. on this topic: Bah, A., Wegmuller, R., Cerami, C. et al. A double blind randomised controlled trial comparing standard dose of iron supplementation for pregnant women with two screen-and-treat approaches using hepcidin as a biomarker for ready and safe to receive iron. BMC Pregnancy Childbirth 16, 157 (2016). https://doi.org/10.1186/s12884-016-0934-8. Again, a well-developed and written paper and I congratulate the researchers involved.
Response 3:
Thank you for your comment. Indeed, our study did not address the effects of iron supplementation on hepcidin during pregnancy. I think a regular use of Fe supplementation will have a profound effect on hepcidin levels, which will affect the relationship between food/nutrients and hepcidin. In this study, the rate of Fe supplementation in relatively low and evenly distributed in hepcidin quartile (Table 1) (p=0.988) (Page 6, Line 252). The low rate of Fe supplementation together with the increasing trend of ID/IDA throughout pregnancy is a worry sign too. Indeed, the Taiwanese government is thinking about providing a Fe supplement for pregnant women who are living in regions with high prevalence of ID/IDA (eastern region of Taiwan). However, the concern is on the recommended Fe dosage, as the recommended dosage by the WHO is very high, which may increase the risk of GI discomfort).
Table 1. Maternal baseline characteristics according to quartiles of serum hepcidin levels (N=1430).
Variable |
Serum hepcidin level, quartiles |
p for trend a |
||||
Q1 (n=355) |
Q2 (n=360) |
Q3 (n=358) |
Q4 (n=357) |
|
||
Basic characteristic………….. |
|
|
|
|
|
|
Reported use of prenatal dietary supplements |
||||||
|
Total supplement use (n, %) |
259 (73.4) |
291 (82.7) |
320 (90.4) |
321 (90.9) |
<0.001 |
|
Multivitamin-mineral (n, %) |
199 (56.5) |
220 (63.2) |
246 (69.5) |
216 (61.4) |
0.071 |
|
Vitamin B (n, %) |
47 (13.5) |
61 (17.4) |
74 (21.1) |
73 (20.9) |
0.005 |
|
Folate (n, %) |
138 (39.2) |
145 (41.4) |
165 (47.3) |
195 (55.7) |
<0.001 |
|
Iron (n, %) |
33 (9.5) |
44 (12.6) |
33 (9.3) |
37 (10.6) |
0.988 |
Blood biochemistry…… |
|
|
|
|
|

Round 2
Reviewer 1 Report
The authors have been responsive to reviewer comments